# Oxidative Stress and Male Fertility: Role of Antioxidants and Inositols

**DOI:** 10.3390/antiox10081283

**Published:** 2021-08-13

**Authors:** Maria Nunzia De Luca, Marisa Colone, Riccardo Gambioli, Annarita Stringaro, Vittorio Unfer

**Affiliations:** 1The Experts Group on Inositol in Basic and Clinical Research (EGOI), 00161 Rome, Italy; m.nunziadeluca@gmail.com (M.N.D.L.); gambioli.riccardo@gmail.com (R.G.); vunfer@gmail.com (V.U.); 2System Biology Group Lab, 00161 Rome, Italy; 3National Center for Drug Research and Evaluation, Istituto Superiore di Sanità, Viale Regina Elena 299, 00161 Rome, Italy; marisa.colone@iss.it

**Keywords:** male fertility, ROS, oxidative stress, antioxidant, inositols

## Abstract

Infertility is defined as a couple’s inability to conceive after at least one year of regular unprotected intercourse. This condition has become a global health problem affecting approximately 187 million couples worldwide and about half of the cases are attributable to male factors. Oxidative stress is a common reason for several conditions associated with male infertility. High levels of reactive oxygen species (ROS) impair sperm quality by decreasing motility and increasing the oxidation of DNA, of protein and of lipids. Multi-antioxidant supplementation is considered effective for male fertility parameters due to the synergistic effects of antioxidants. Most of them act by decreasing ROS concentration, thus improving sperm quality. In addition, other natural molecules, myo-inositol (MI) and d-chiro–inositol (DCI), ameliorate sperm quality. In sperm cells, MI is involved in many transduction mechanisms that regulate cytoplasmic calcium levels, capacitation and mitochondrial function. On the other hand, DCI is involved in the downregulation of steroidogenic enzyme aromatase, which produces testosterone. In this review, we analyze the processes involving oxidative stress in male fertility and the mechanisms of action of different molecules.

## 1. Introduction

Infertility is diagnosed after at least one year of regular unprotected intercourse without conception. Diagnosis of infertility has become a global health concern, occurring in about 187 million couples worldwide, and approximately half of the cases are attributable to male factors [1,2].

Oxidative stress caused by a high amount of reactive oxygen species (ROS) has been observed in 30–80% of infertile patients [3,4]. High levels of ROS promote impairment of sperm quality mainly by decreasing motility and increasing the levels of DNA oxidation, protein oxidation and lipid peroxidation [5].

Lifestyle [6,7], genetics [8] and environment, such as exposure to chemicals [9], represent risk factors for male infertility. Additionally, several pathologies may cause infertility, including varicocele and endocrine unbalance, all related to oxidative stress and DNA damage [6].

Infectious diseases represent another important factor that may affect human fertility, especially through damage to DNA, inflammation and oxidative stress. In particular, viruses are known to induce inflammation in infected tissues, promoting the production of ROS [10]. A notable case of virus infecting testes, among others, is the SARS-CoV2. Such a virus is internalized by the cells via angiotensin-converting enzyme 2 (ACE2), which is highly expressed on testis, especially in Leydig and Sertoli cells [11]. Clinical studies report that SARS-CoV2 infection increases the concentration of ROS, together with malondialdehyde (MDA). Moreover, such a virus seems to reduce the percentage of motile spermatozoan, impairing both motility and morphology [12,13].

Indeed, ROS represent a major factor that contributes to male idiopathic infertility. In recent years, male oxidative stress infertility (MOSI) emerged as a term defining “infertile men with abnormal semen characteristics and oxidative stress” that has a global incidence of about 37.2 million cases. MOSI is characterized by abnormal semen parameters with no clear cause of idiopathic infertility [14].

The oxidative stress is an emerging risk factor for male infertility, and antioxidants are recommended as treatment of choice for idiopathic infertility. Intriguingly, different antioxidants display synergistic effects, making multi-antioxidants an effective treatment for male infertility [14]. Treatments for male infertility involve several different compounds. Most of them act by decreasing the levels of ROS and thus improving sperm motility [15].

In this regard, myo-inositol is an important natural compound whose action as an antioxidant molecule is well documented. The structure of inositols is composed by a ring made out of six carbons, each with a hydroxyl group substituent. Myo-inositol (MI) is the most common stereoisomer. Being MI a fundamental element of membranes, it is involved into osmoregulation and protein phosphorylation. Phosphorylated forms of MI are second messengers of different pathways. Likewise, in sperm cells MI participates in transduction mechanisms that control calcium levels in cytoplasm, other than capacitation and mitochondrial functionality [16].

Recently, the optimal dosages and timing for the administration of D-chiro-inositol (DCI) was evaluated for different pathologies, including male infertility. Indeed, the evaluation considered the effect of DCI on the expression of steroidogenic enzyme aromatase, which aromatize androgens to estrogens. Specifically, treatment with DCI inhibits the expression of the enzyme, leading to testosterone accumulation [17].

## 2. Dual Role of ROS and Antioxidant System in Male Fertility

Oxidative stress is an important cause of male infertility due to detrimental changes during spermatogenesis, epididymal maturation, and sperm capacitation, that can lead to infertility [18].

Spermatozoa are produced in the testes during the hormone-regulated process of spermatogenesis. The crucial step to achieve fertilization capacity, motility and complete maturation occurs into epididymis [18,19]. During this phase, spermatozoa are physiologically exposed to ROS that are also involved into physiological functions such as sperm capacitation and acrosome reaction. These are necessary for efficient fertilization and require high levels of energy provided by metabolic pathways as glycolysis or oxidative phosphorylation (OXPHOS). Capacitation is a cascade of different cellular reactions that enables spermatozoa to bind the zona pellucida of oocyte. This induces the acrosome reaction, a release of proteolytic enzymes [18].

Another feature to consider is that the sperm membranes are made up of a high amount of polyunsaturated fatty acids (PUFA), which guarantee the fluidity necessary for fertilization. At the same time, this high amount of PUFA represents a risk for the spermatozoa, being PUFA vulnerable to lipid peroxidation (LPO) [19]. The oxidative damage that can result is associated with the loss of membrane fluidity, mitochondrial dysfunction, alteration of morphology, reduction of vitality and other alterations that lead to the failure of fertilization [6].

Furthermore, the structure of mature spermatozoa has no capacity to respond to stress stimuli because the haploid and highly compacted nucleus does not transcribe anymore. Indeed, during the last stages of spermatogenesis, spermatozoa has eliminated most of its cytoplasm, which is the major source of antioxidants [20]. So, in spermatozoa, the cytoplasm content is very small and it is mainly occupied by DNA [19].

While physiological levels of ROS are necessary for the regulation of spermatic functions, an excessive quantity can overwhelm the antioxidant mechanisms responsible for the protection of spermatozoa. In the seminal fluid there is an antioxidant system dedicated to maintaining normal cellular function, composed of enzymatic and non-enzymatic factors, which interact with each other to ensure optimal protection against ROS. Among these factors, important roles are covered from the enzymatic triad that includes superoxide dismutase (SOD), catalase (CT) and glutathione peroxidase (GSHPX) [18,21]. 

SOD is a metalloenzyme that catalyzes the superoxide anion dismutation reactions and specifically, plays a leading role in the protection of PUFA, constituents of the plasma membrane, and in the fragmentation of DNA [22]. It can perform its functions both in extra and intracellular space, even though the principal enzymatic activity is located to the cytoplasm of the cells [23].

Catalase (CT) is responsible for the transformation of hydrogen peroxide into molecular oxygen and water. It is characterized by the presence of the heme system, with an iron atom in the center of the group. Its activity has been found in different organelles: peroxisomes, mitochondria, endoplasmic reticulum and in the cytosol of different cell types [23]. 

Another enzyme that is part of the sperm antioxidant system is glutathione peroxidase (GSHPX), whose active site is made up of selenocysteine. Generally, this enzyme is responsible for the reduction of hydrogen peroxide and organic peroxides [23]. There are three isoforms: cytosolic, mitochondrial and nuclear. Specifically, the mitochondrial isoform is necessary for sperm quality and motility [22]. 

When the antioxidant system fails to counteract the excessive increase in ROS, cell death occurs. Therefore, ROS have a double role: physiologically to complete the maturation of spermatozoa and/or to achieve fertilization, but in excess they are harmful to cell structures, functions and survival [6].

### Genesis of Oxidative Stress, Lipid Peroxidation and DNA Damage

When ROS production passes antioxidant defenses, dangerous effects on spermatozoa can be summarized as increased LPO and DNA damage and reduction of sperm motility, morphology and viability which are associated with lower sperm fertility [23]. The main sources of ROS are attributable to two metabolic pathways that produce energy: glycolysis and OXPHOS [19].

To produce ATP molecules, the cells rely on mitochondria, which generate ROS during mitochondrial respiration. [24]. The electron transport chain and oxidative phosphorylation generate ATP in the mitochondria, transferring electrons from the inner mitochondrial membrane complexes. Consequently, the process leads to a pumping of protons to the intermembrane space. Here, the electron chain may generate a local excess of ROS, especially by complexes I and III [23].

The production of ROS starts with the formation of a superoxide anion radical (O_2_^−^), which is physiologically recycled by SOD into hydrogen peroxide (H_2_O_2_). Hydrogen peroxide is very stable molecule and can cross the plasma membrane to contact all the cellular and extracellular compartments. Despite that hydrogen peroxide is one of the non-radical species, it can generate highly reactive hydroxyl radicals in the presence of metal ions. Via the Fenton and Haber–Weiss reactions, an excess of H_2_O_2_ leads to the generation of very aggressive radicals (hydroxyl and alkoxyl, OH and OH^−^).

Furthermore, the main target of ROS is the peroxidation of unsaturated fatty acids that produces highly reactive lipid and aldehyde, which in a positive feedback react with cellular components producing increasing ROS contents [6,25].

Spermatozoa are also particularly susceptible to the damage induced by excessive ROS because their plasma membranes contain large quantities of PUFA as the decohexaenoic acid (DHA, where six double bonds between their methylene groups are not conjugated). PUFA undergo lipid peroxidation by ROS, and this reduces the integrity of the membranes [26].

LPO is divided in three subsequent phases, schematized in Figure 1:Initiation: corresponds to the extraction of hydrogen atoms from the carbon-carbon double bonds of an unsaturated fatty acid to generate free radicals;Propagation: corresponds to the formation of lipid radicals followed by their rapid reaction with oxygen to form peroxyl radicals;Termination: corresponds to the last phase in which the radicals formed react with other lipids generating different cytotoxic adducts such as aldehydes [25].

As previously mentioned, excess ROS can damage not only the fluidity of the sperm membrane but also the nuclear DNA. Damage related to sperm DNA can result in an attack on the nitrogenous bases, a double or single strand break of DNA, and chromatin alterations [21]. Previous studies conducted on the oxidative stress of spermatic DNA have shown that guanosine and adenosine are among the nucleosides most sensitive to oxidation [25]. It is widely known that genetic material is structured in a highly condensed and compact manner to increase stability and at the appropriate time DNA is decondensed to transfer genetic information [21]. In this regard, the excess of ROS can promote decondensation by exposing the DNA to the damage of free radicals (DNA repair can only occur during specific stages of spermiogenesis which does not include the condensation stage. In fact, after the initial stages the last chance for repair occurs by the human oocyte. If the breaks in the filaments are not repaired, the cell undergoes apoptosis and therefore programed cell death [25].

## 3. Role of Antioxidant in Male Fertility

Researchers observed an enhancement in semen parameters with the use of antioxidants, suggesting that such substances minimize the toxic effects of oxidative stress in spermatozoa.

### 3.1. Folic Acid

One of these antioxidants is folate, a vitamin from the B group involved in many biochemical processes and several functions as: DNA synthesis, which is fundamental for the development of spermatozoa; oxidative pathway, as the synthetic form of the folate, folic acid, effectively scavenges oxidizing free radicals and inhibits LPO [27,28,29]. Different studies evaluated the free radical scavenging properties and possible antioxidant activity of folic acid. Its constituents, pyrazine and pterin, can easily be reduced by hydrated electron to the corresponding hydroderivatives in the pyrazine ring of the molecule [30,31]. Furthermore, free radical intermediates are suggested in the chemical oxidation of reduced pterins by air, H_2_O_2_ or iron ions [32,33]. Moreover, a folate deficiency is involved into apoptosis process through p53, as it happens in case of certain types of DNA damage [34]. As a consequence of the involvement of folate in scavenging processes, researchers found that the concentration of indexes of lipid peroxidation in folate-deficient cells are drastically increased. This folate deficiency activates a redox-sensitive transcription factor, NF-κB, which controls an apoptosis mediated by reactive oxygen species [35].

Different studies analyzed the supplementation of folic acid in sub-fertile male. A study reported an improvement in number and motility of spermatozoa and a decrease in the number of immature cells after 3 months of supplementation with 15 mg of folic acid (5-formyl tetrahydrofolate) in 65 men of infertile couples with cell idiopathic syndrome [36]. A recent systematic review and meta-analysis analyzed seven randomized controlled trial (RCT) involving sub fertile men to evaluate oral folic acid supplementation alone or in combination with zinc sulfate, evaluating inhibin B, FSH, testosterone and concomitantly sperm characteristics as concentration, morphology and motility. Folic acid may also improve endocrine parameters by stimulating the Sertoli cells, the main producers of inhibin B. The serum concentration of inhibin B relates with sperm concentration, testicular volume and the state of the spermatogenetic epithelium. Intuitively, the concentration of inhibin B reflects the quality of the Sertoli cell and thus represents a marker of good spermatogenesis in humans [37,38]. As a consequence, supplementation with folate significantly improve sperm concentration [39,40].

### 3.2. L-carnitine

L-carnitine is detectable as free or acetylated forms in epididymal tissue, seminal plasma and spermatozoa [41,42]. The pivotal role of L-carnitine is to transport acetyl and acyl groups, which are essential for mitochondrial metabolism, across the mitochondrial inner membrane. L-carnitine likely accelerates the metabolism of long-chain fatty acids in mitochondria [42,43]. During this process L-carnitine temporarily binds acetyl groups, producing L-acetyl-carnitine [42]. These reactions modulate mitochondrial concentrations of acetyl coenzyme A (CoA), which is implicated in the energetic metabolism, such as the Krebs cycle, the β-oxidation of organic acids and the degradation of amino acids [43]. L-acetyl-carnitineand L-carnitine participate in the energetic metabolism, which improves the motility and the maturation of spermatozoa. Furthermore, L-carnitine and L-acetyl-carnitine participate in protection against oxidative damages. The excess of acetyl-CoA generated allows the formation of L-acetyl-carnitine as a buffer of acetyl groups [44]. Moreover, L-carnitine shows the activity of free radical scavenger, especially to superoxide anion. In a study involving rats, L-carnitine taken before doxorubicin, a chemotherapy drug, partially preserved the acrosome integrity of sperm [45]. In another study on mice, L-carnitine raises Bcl-2 levels and reduces Bax expression, indicating that this compound may inhibit apoptosis [46]. Several studies on L-carnitine involved also antimicrobials, anti-inflammatory drugs or pentoxifylline [47,48,49,50]. These studies showed positive results as an increase in sperm vitality, motility and a reduction in ROS. The treatment also improved the sperm count when in combination with pentoxifylline [51].

A randomized placebo-controlled study on 100 patients taking L-carnitine showed a significant improvement in semen quality, specifically in sperm concentration and motility [52]. The same author conducted another randomized, placebo-controlled study from the same group involved 60 OAT patients taking either L-carnitine, L-acetyl-carnitine or a combination. The combined treatment improved sperm motility, especially in groups with lower levels at baseline [53]. A further study confirmed that L-acetyl-carnitine improved sperm motility, either alone or in combination with L-carnitine. The combined therapy significantly improved straight progressive velocity after 3 months [54]. A double-blind randomized cross-over clinical trial on 30 infertile men evaluated the treatment with L-carnitine or placebo. After a washout period of 8 weeks, each group received the different treatment. This posology significantly improved sperm concentration and progressive motility [55]. A further study on 20 patients with idiopathic oligoasthenospermia taking L-acetyl-carnitine significantly increased progressive sperm motility [56]. A multicenter study on 100 patients receiving 3 g/day of oral L-carnitine for 4 months indicates that L-carnitine increases spermatozoa motility and the total number of ejaculated spermatozoa [57]. A single-blind clinical study on 30 asthenozoospermic patients taking 2 g/day of L-carnitine for 3 months revealed an improvement of mean sperm motility only in patients with normal GSHPX levels. Since GSHPX plays pivotal role in male fertility, L-carnitine treatment might improve sperm motility in the presence of normal mitochondrial function [58]. Ultimately, a systematic review quantified the efficacy of L-carnitine and/or L-acetyl-carnitine. Results showed that both carnitines likely improve total sperm motility and reduce the percentage of sperm with incorrect morphologies [59].

### 3.3. L-arginine

L-arginine actively participates in the formation of sperm and prevents the peroxidation of membrane lipids [60]. This mechanism seems to involve nitric oxide (NO), a short-lived free radical, synthesized in many mammalians cell types by a class of NADPH dependent enzymes called nitric oxide synthases (NOS) [61]. These enzymes catalyze the conversion of L-arginine to L-citrulline and NO [62]. In vitro studies investigated the effects of exogenous NO donors on sperm function as motility and viability, with controversial results. There is evidence that low concentrations of NO increase human sperm capacitation [63]. In addition, other studies suggest that the stimulation of NO generation relates with the enhancement of tyrosine phosphorylation in sperm proteins, which leads to sperm capacitation [64]. Moreover, NO inactivates superoxide anions. When NO predominates, it inactivates superoxide; when superoxide predominates, it inactivates NO. Thus, higher concentration of NO is expected to reduce lipid peroxidation by inactivating superoxide. Based on the ability of L-arginine to increase the generation of NO, it is likely that L-arginine protects spermatozoa against lipid peroxidation. These results strongly support the proposal that L-arginine improves the quality of spermatozoa via the biosynthesis of nitric oxide [65].

Unlike the other substances aforementioned, few studies exist on L-arginine and sperm parameters. In a first in vitro study, human semen with low motility incubated with L-Arginine showed that the amino-acid enhances sperm motility. This suggests that L-arginine may be a useful treatment in artificial insemination processes in men with subnormal spermatozoa motility [66]. In a goat epididymal spermatozoa L-arginine plays an important role in its physiology and it enhances the metabolism of these cells. It also decreases the grade of membrane lipid peroxidation. The authors evaluated both natural peroxidation and that induced by UV radiation. Independently from the nature of peroxidation, L-arginine reduces the extent of lipid peroxidation in a concentration dependent manner [67]. A further study investigated the clinical efficacy and acceptance of L-arginine in 40 infertile men. All of these men had a normal number of spermatozoa (>20 million/mL), but a decreased motility. They received 80 mL of 10% L-arginine HCL daily per os for 6 months. L-arginine treatment led to an improvement in the motility of spermatozoa without any side-effects [68]. In another study, 45 patients with various degrees of oligospermia and asthenospermia received L-arginine, indomethacin and kallikrein. Fifteen patients took L-arginine hydrochloride, 15 the anti-inflammatory agent indomethacin, while 15 others the enzyme kallikrein, all for 3 months. All the three treatments increased sperm count and motility [69]. In a further double-blind, randomized, placebo-controlled, crossover study, physicians analyzed the sperm quality of 50 subfertile men after the treatment with 2 tablets containing 3 g L-arginine aspartate or placebo for 1 month. The amino-acid improved sperm volume, concentration, motility, vitality and morphology without adverse effects [70].

### 3.4. N-acetylcystenine

Since the 1960s, N-acetyl-cysteine (NAC), has been widely described as a mucolytic agent. In particular, the mucolytic action of NAC is due to its ability to break the disulfide bonds in the high-molecular-weight glycoproteins of mucus, reducing the viscosity. For this reason, NAC is also considered as an option for the treatment of diseases involving oxidative stress. In addition, several in vitro studies reported efficient antioxidant activity of NAC using different oxidants, substrates, and methods to assess the oxidative processes [71,72,73,74,75,76]. The antioxidant activity of NAC can be related to at least three different mechanisms:(1)A direct antioxidant effect toward certain oxidant species including NO_2_ and hypohalous acids (HOX). HOX, due to their high reactivity, are not specific oxidants and also react with many biologically important molecules, thus inducing a cytotoxic effect [77].(2)As NAC acts as a predecessor of cysteine and is part of important step to glutathione synthesis has an indirect antioxidant effect. Then GSH is engaged in different detoxification processes as elimination of by-product of lipid peroxidation and hydroperoxides [78].(3)From a chemical point of view, NAC acts as a reducing agent, and therefore exerts its activity against the disulfide groups by reducing them and generating SH group [79,80].

Further studies involved animal models to evaluate NAC efficacy. For example, a study evaluated the protective effect of NAC against the toxic effects of orally administered TiO_2_ nanoparticles in 50 adult male albino rats. It is known that the TiO_2_ particles trigger pathological alterations at the testicular level, which results in an increase in MDA and a corresponding reduction in GSH. The simultaneous administration of NAC restores the previously mentioned alterations by exerting a protection against DNA damage [81]. Another study on mice evaluated the protective role of NAC against arsenic trioxide (As_2_O_3_), which is often used in treatment of leukemia. Following NAC administration, animals showed improved sperm parameters and seminal vesicle weight [82]. The exposure to another substance, chlorpyrifos (CPF), may cause chronic toxicity in male genital system, and the treatment with NAC after the exposure significantly improves spermatogonia, spermatocytes, spermatid cell counts as well as sperm parameters [83].

In vitro studies evaluated the effect of NAC on human spermatozoa using Stattic V, a non-peptidic inhibitor of STAT3 involved into sperm function [84,85]. Stattic V is associated to sperm apoptosis and sperm immobilization due to increase of different pathological outcomes such as acrosome reaction, intracellular Ca^2+^ concentration, extracellular levels of reactive oxygen species (ROS), mitochondrial membrane depolarization and decrease in sperm ATP content [86]. Other in vivo study assessed the effect of 600 mg/day of NAC for three months to evaluate different parameters as chromatin negative alteration induced by high oxidative stress and its consequences on sperm quality (motility, count and morphology). After this treatment all the sperm parameters improved significantly and in parallel DNA fragmentation and protamine deficiency decreased. Positive results were also into hormonal profile, lowering FSH and LH levels e consequently increasing testosterone levels [87].

As the production of reactive oxygen species (ROS) is one of the main events associated with varicocele, physician evaluated the efficiency of NAC supplementation in 35 infertile men with varicocele. The researchers evaluated semen parameters, protamine content, DNA integrity and oxidative stress before varicocelectomy and three months later. Abnormal semen parameters, protamine deficiency, DNA fragmentation and oxidative stress were significantly decreased either in the control or in the treatment group compared to before surgery, and in particular NAC significantly improved protamine deficiency and DNA fragmentation [88].

A further double-blind, placebo-controlled, randomized study included 468 infertile men with idiopathic oligo-asthenoteratospermia. After randomization, patients received either 200 mcg selenium orally daily, 600 mg N-acetylcysteine orally daily, 200 mcg selenium plus 600 mg N-acetyl-cysteine orally daily or similar regimen of placebo for 26 weeks. In response to selenium and N-acetyl-cysteine treatment, serum FSH decreased but serum testosterone and inhibin B increased. All semen parameters (concentrations, motility and percent normal morphology) significantly improved with selenium [89].

Another in vivo study included 120 men with idiopathic infertility, divided randomly into 2 groups: the first treated with NAC (600 mg/day orally) for 3 months and the second with placebo. After NAC treatment, the serum total antioxidant capacity was greater and the total peroxide and oxidative stress index were lower in the in respect to the control group. These beneficial effects resulted from reduced reactive oxygen species in the serum and reduced viscosity of the semen [90].

Another randomized, blinded clinical trial study on 50 asthenoteratozoospermic men evaluated nuclear factor erythroid 2-related factor 2 (NRF2). NRF2 activates the cellular antioxidant response by inducing the transcription of a wide array of genes that can combat the harmful effects of factors such as oxidative stress. After the treatment with NAC (600 mg, three times daily), researchers found a significant increase in sperm concentration and motility compared to pre-treatment status, whereas the percentage of abnormal morphology and DNA fragmentation was significantly decreased. The authors also observed a significant improvement in the expression of NRF2 gene and in antioxidant enzyme levels. They also report significant correlations between NRF2 mRNA expression level, specific sperm parameters and level of antioxidant enzymes [91].

### 3.5. Resveratrol

Resveratrol (RSV) is a natural polyphenolic compound, found primarily in grapes and wine, presenting a considerable number of beneficial effects in a variety of organs and systems, which are mainly due to its antioxidant activity [92]. RSV inhibits the formation of ROS via the suppression of pro-oxidant genes and the induction of antioxidant enzymes including SOD, CAT, and GSHPX [93,94,95,96,97]. RSV also chelates copper and other transition metals, which are able to generate free radicals and thus may cause lipid peroxidation [98].

RSV exerts a protective action on the testes and epididymis, as it acts as an antioxidant and a scavenger of electron-donor free radical [99,100]. RSV is a lipophilic molecule that prevents lipid peroxidation induced by Fenton reaction products and its protective effects against oxidative damage is likely due to a hydrogen electron donation from its hydroxyl groups [101,102,103]. Moreover, RSV inhibits phosphodiesterase enzymes (PDE), and this action leads to an increase in intracellular cyclic adenosine-monophosphate (cAMP) [104]. Such a mechanism is of particular relevance to mammalian spermatozoa, as the majority of the processes involved in their capacitation are modulated by a cAMP-dependent signaling cascade [105].

Another mechanism of action could be related to AMPK pathway. AMPK activation has been reported to reduce ROS levels in different animal models and pathology and also upon human and chicken sperm cryopreservation. Protective features of RSV on cryopreservation-induced oxidative stress may be mediated through activation of AMPK [106].

Several studies evaluated the capacity of RSV to protect from DNA damage in cryopreservation of human semen. The addition of resveratrol before the cryopreservation process avoided oxidative damages to the sperm, in both fertile and infertile men [107].

In a study the protective effect of resveratrol was evaluated against polyvinyl chloride (PVC), used in the plastic industry, to evaluate its toxicity on male fertility measuring oxidative status and specifically evaluating steroidogenesis, spermatogenesis, in an animal preclinical model with adult male Wistar Rats. RSV is transported into the mitochondria by StAR at the hydrophobic tunnel and can reduce oxidative stress induced by PVC protecting mitochondria from oxidative damage and consequently preserving the physiology of Leydig cells for testosterone biosynthesis [108]. Additionally, acrylamide was tested and administered to male 405 mice for three or six months. The use of RSV, as anticipated in other results also reported above, improves the damage to the DNA level. However, it is necessary to underline that the treatment for six months also brought negative results such as alteration of the morphology, probably caused by a premature induction of the capacitation process [109].

In another preclinical study the capacity of RSV was assessed to contrast the ferrous iron/ascorbate damage in sperm mice. The treatment with RSV provided before ferrous iron/ascorbate treatment showed a significant increase of MMP parallel to significant decrease of ROS and consequently positive results to sperm parameters such as viability and motility. However, no changes in SOD activity were observed. To evaluate the effect of valproic acid (VPA), a drug widely use for the treatment of epilepsy in male reproductive function, Wistar rats were treated with VPA by gavage for 28 days in co-administration of RSV. RSV was shown to minimize the damage induced by VPA and protect the male reproductive system. The results reported a strengthening of antioxidant capacity both in testes and in the epididymides of treated rats that were previous treated with long VPA treatment [110]. Another pathological situation that can induce lipid peroxidation and therefore increase oxidative stress and damage related to fertility is type 1 diabetes mellitus (DM1). RSV has been shown to be useful in improving the parameters of fertility altered by this pathology: lipid peroxidation, fragmentation of spermatic DNA, alteration of chromatin and mitochondrial mass [111].

## 4. Role of Inositols in Male Fertility

Inositol exists as nine stereoisomers, resulting from epimerization of the six OH- groups (cis-, epi-, allo-, myo-, neo-, scyllo-, L-chiro-, D-chiro-, and muco-inositol). The most diffused form in nature is cis-1,2,3,5-trans-4,6-cyclohexanehexol, or myo-inositol (MI), followed by D-chiro-inositol (DCI) [112,113,114]. The conversion of MI into DCI is accomplished by an enzyme with epimerase activity, which is responsible for their different tissue distribution. In fact, every organ or tissue has a specific ratio of MI:DCI, fundamental for the correct inositol-related functions. In animals, inositol is both synthesized and taken with the diet through grains, wheat germ, citrus fruits and meats, primarily as inositol-phosphates [115,116,117,118]. The absorption of free inositol in the intestine is achieved with an active transporter of sodium/myo-inositol transporter (SMIT) family, temperature and pH dependent. SMIT is an active symporter which allows uptake and accumulation, dependently on both concentration of substrates and energy [119,120]. Furthermore, intestinal absorption of a high amount of myo-inositol is considered safe, as highlighted by several studies that proved the absence of side effects even after ingesting large quantities [118]. In addition, in vitro and in vivo results support an increased MI intestinal absorption when combined with alpha-lactalbumin administration [121]. Once absorbed, inositol-phosphates are metabolized into un-phosphate MI by inositol-phosphate-phosphatase 1 (MINPP1) and then transported to the cytosol of cells. Specifically, an integral membrane protein of the SMIT family encoded by the SLC5A3 gene and controlled by osmoregulatory elements transports one molecule of MI and two of Na^+^ [122,123,124]. In recent years, the SLC5A3 cellular transporter was found to be expressed in several tissues. Among others, testis and epididymis displayed (high/medium/low) expression, while Sertoli cells display elevated expression, as the hypertonic conditions increase MI uptake [119,120]. Therefore, here the concentration of MI is 28 times higher compared to the plasma. Moreover, the presence of the blood-testicular barrier prevents the free passage of MI from the blood to the testicle and vice-versa. Owing to this mechanical barrier, MI remains strongly concentrated into seminiferous tubules [120]. In spermatozoa, MI plays a key role as an intracellular second messenger through the regulation of Ca^2+^ levels. It also intervenes in the regulation of sperm motility, capacitation and the acrosome reaction. The activation of intracellular transmission systems necessarily leads to an increase in cytoplasmic and mitochondrial Ca^2+^ level. High Ca^2+^ levels stimulate the oxidative metabolism, inducing the production of ATP depending on the energy requirements [124]. This system needs a good functional state of the mitochondria and consequently of high mitochondrial membrane potential (MMP). Recently, researchers proved that high MMP correlates to higher fertilizing capacity of spermatozoa and to higher spermatic motility. Several studies highlighted an interesting role of MI, which proved able to increase MMP and sperm motility [125]. Finally, regulating intracellular levels of Ca^2+^, MI is able to improve the main characteristics related to the “state of health” of spermatozoa, increasing their fertilizing capacities [126]. The acrosome reaction is a prerequisite for fertilization in mammals. It consists of a fusion between the plasma membrane and the outer acrosome membrane above the anterior portion of the sperm head. It takes place on the surface of the zona pellucida, after specific binding with a specific glycoprotein, the ZP3 [127]. Once the acrosome reaction is completed, the sperm cell is able to penetrate the zona pellucida. A premature acrosome reaction leads to a loss of the recognition sites of the zona pellucida on the spermatozoa surface and thus it affects gamete fusion [128]. In contrast, the inability to achieve activation, which is responsible for initiation of the acrosome reaction, prevents the oocyte penetration.

Moreover, in IP3 form, MI is involved in the activation of Akt, which is a fundamental protein involved in maturation of spermatozoa. In fact, Akt regulates a wide range of proteins by phosphorylation [129]. Akt phosphorylating some proteins as Bcl-2 (at the level of the residue Ser70, S473, T308) negatively regulates the apoptosis process. Indeed, phosphorylation determines the passage from the mitochondria to the cytosol and making it a marker of cell survival [130,131]. The MI in the IP3 form initiates a cascade of reactions which always lead to the phosphorylation of tyrosine residues and reflect the state of capacitation that makes male gamete available to fusion with the oocyte (acrosome reaction) [132,133,134].

MI also represents the second messenger of the gonadotropin Follicle-stimulating hormone (FSH). At the testicular level, FSH plays a key role in the control of Sertoli cell number and function, promoting the differentiation of these cells essential to sustain a normal spermatogenesis. MI, acting as second messenger, regulates the activities of FSH, and thus may result useful to counteract alterations of the hormone levels. In addition, MI administration lowers LH concentration. High FSH and LH serum concentrations relate with low sperm concentration [135,136,137,138,139]. On the other hand, MI increases the levels of Inhibin B, a glycoprotein secreted from the testis as a product of Sertoli cells that act as negative feedback to regulate FSH secretion. In men, either with normal or altered spermatogenesis, a strong inverse correlation is reported between inhibin B and FSH levels [140,141].

### 4.1. MI: In Vitro Studies

One of the first studies evaluating the impact of MI treatment on sperm samples from OAT patients showed that 2 mg/mL MI determine the absence of amorphous material. This material is responsible for the high viscosity of the seminal fluid and consequently for the reduction of sperm motility. Furthermore, the mitochondria of the treated cells showed a morphology similar to the physiological, free from damage to the mitochondrial crests, in contrast to the untreated ones, which display altered morphology [142]. To investigate the effects of MI on mitochondrial function, researchers incubated samples from OAT patients with 2 mg/mL and then evaluated: MMP, phosphatidylserine externalization (PS) and chromatin compactness. At the end of the treatment, although there were no appreciable results on PS and chromatin compactness, the number of spermatozoa with high MMP had increased, otherwise the number of spermatozoa with low MMP had decreased [143]. Another similar in vitro study showed the same outcomes of higher sperm numbers with high MMP and higher progressive motility in both normospermic and OAT patients. Furthermore, motility improvement in the first group was associated with a significant increase in the percentage of spermatozoa with high MPP [144]. As studies showed that sperm motility is directly associated to fertilization rate, even in IVF procedures, different studies evaluated the impact of MI in IVF procedures [145,146]. For this reason, MI use was shown to improve the culture conditions necessary for a successful ICSI technique. This led to improvement in outcomes as the fertilization rate, the percentage of grade A embryos on day 3 and progressive motility in normospermic and OAT patients undergoing in vitro fertilization (IVF) [147,148]. In sperm samples from patients with hyper viscosity, MI also improved progressive motility compared to the control group [149]. As the thawing process of sperm samples leads to a reduction in motility, sperm quality and fertilization rate, the efficacy of MI was evaluated on both fresh and thawed sperm samples. The results obtained showed an improvement in motility in both samples [150]. Similar results were obtained when evaluating the use of MI in culture media for cryopreservation processes. Again, the results showed a significant increase in the cryo-survival rate (CSR), defined as the percentage of total motility after thawing, divided by the percentage of total pre-freezing motility and multiplied by 100 [151,152]. Hence, in vitro supplementation of MI has been shown to induce a significant increase in sperm motility and oxygen consumption, the main index of the efficiency of oxidative phosphorylation and ATP production. Previous studies provided only indirect results on the antioxidant power of MI. Therefore, researchers decided to evaluate 8-OHdG, one of the first products of oxidative damage to DNA. The samples treated with MI displayed reduced levels of 8-OHdG [153]. Therefore, MI can improve the sperm parameters related to the quality and in vitro fertilization process. In fact, the antioxidant properties of MI, although not yet fully known, can contribute to the improvement of sperm parameters to optimize the results of assisted reproduction techniques. Data from in vitro studies are collected into Table 1.

### 4.2. MI: In Vivo Studies

Several studies evaluated the improvements in sperm and metabolic parameters related to male infertility following dietary supplement based on MI. A study carried out on patients with idiopathic infertility demonstrated that MI can be useful for improving sperm parameters such as percentage of spermatozoa with acrosome reaction, spermatozoa concentration, total count and progressive motility, when compared to placebo. Furthermore, the same study evaluated hormonal parameters and highlighted a reduction in follicle-stimulating hormone and luteinizing hormone and a concomitant increase in inhibin B concentrations [154]. In another study, samples from healthy and oligoasthenospermic (OA) patients were analyzed by light microscopy to evaluate semen volume, sperm number and motility before and after the density gradient separation method. The study considered these parameters before and after the administration of 4000 mg/day of MI and 400 mg of folic acid for 2 months. After treatment there was a significant increase in sperm concentration in the OA patient group and a significant increase in sperm count in the healthy patient group [155]. Furthermore, such supplementation highlighted promising results in asthenospermic patients with metabolic syndrome, showing significant improvements in sperm (concentration, motility and morphology), hormonal (testosterone, E2, LH, SHBG) and metabolic (HOMA index) parameters [156]. In addition, 85.32% of asthenospermic patients achieved significant improvement in sperm motility. In particular, 34.86% of patients restored normal sperm motility while only 12.84% showed no beneficial effect [157]. For the first time, researchers investigated the effect of MI on cholesterol efflux, a hallmark of capacitation. They underlined an increase in cholesterol efflux in the spermatozoa of patients with OAT treated either in vitro or in vivo with a blend of nutraceuticals, containing mainly MI. The same study also found an increase in the activity of G6PDH, associated with the increase in glucose metabolism through pentose phosphate pathway (PPP), both in normal patients and in patients with OAT [158]. Data from in vivo studies are collected into Table 2.

### 4.3. DCI in Male Fertility

Profound differences between the physiological functions of MI and DCI exists, and physicians should pay attention on their use in assisted reproductive treatments. Indeed, it is reported that the 40:1 ratio supplementation of MI and DCI in the treatment of PCOS is one of the most effective [159,160].

Particularly, several studies indicate that the inositol plays a crucial role in oocyte and spermatozoa development. An imbalance between MI and DCI may lead to a reduction in insulin and FSH signaling, as observed in PCOS patients [161]. In this regard, administration of MI, alone or in combination with DCI (in the physiological plasma ratio of 40:1), could be an adjuvant factor in improving ART outcomes [162].

Regards to mitochondrial function, and its membrane potential, MI showed different positive results on correlated parameters as motility, thanks to insulin-sensitizing, antioxidant, prokinetic, and hormonal properties. Although the DCI showed positive results on sperm mitochondrial function in vitro, MI also plays a crucial role in the development of oocytes, spermatozoa and the embryo. In fact, for this reason MI is used in medically assisted reproduction techniques, both for the male and female factor [129,159].

On the other hand, as MI, DCI acts as insulin sensitizer and is incorporated within the inositol-phosphoglycans (IPG) as second messengers of insulin. In particular, DCI mediates glycogen synthesis and stimulates androgen production at the ovarian level [163]. In fact, DCI concentrations are higher in the tissues responsible for glycogen storage (liver, muscle, fat) and lower in the tissues with high glucose utilization (brain, heart, ovaries) [164]. Further studies showed that DCI reduces the gene expression of aromatase in a dose-dependent manner [165]. Aromatase is an enzyme present in different tissues that synthetizes estrogens from androgens, and modulating its gene expression regulates free and total testosterone levels [166]. In this context, DCI can be widely used in those pathologies characterized by an increase in estrogen levels or a lowering of androgen levels, such as in several contexts of male infertility. In particular, DCI can be useful in clinical pictures related to alterations of sperm parameters associated with low testosterone levels [167].

Several aromatase inhibitors (AIs) were studied in male fertility, as they can be useful to rebalance T/E2 ratio, restoring also the hypothalamic-pituitary-testis axis and thus the resulting spermatic alterations. In case of no alteration of hormonal component, the administration of AIs generally leads to fewer results on sperm alterations. An excess of testosterone at testicular level can also have harmful effects on spermatogenesis and an excessive reduction in E2 levels could excessively block the negative feedback at the level of hypothalamic-pituitary-testis [168,169]. In addition, in the last years, the role of a discrete amount of E2 at testicular level emerged as a component strictly required for development of germ cells [170,171]. In this regard, a recent study highlighted that DCI could be a suitable treatment in hypogonadal hypogonadotropic patients, who display lowered androgen levels due to the ageing. In fact, inhibiting aromatase expression and naturally restoring physiological androgen levels revert the pre-pathological hypogonadism condition. On the other hand, such as aromatase inhibitors, DCI must be avoided in primary hypogonadal males, where it could worsen the already overburdened gonadotropin signaling.

## 5. Conclusions

Even though antioxidants exert protective roles against ROS-induced damage, they do not always represent the most suitable solution in case of ROS-independent infertility. Nevertheless, antioxidant could represent an adjuvant treatment in such cases, avoiding further damage caused by ROS. Inositols play also other roles than those in the antioxidative pathways. Thus, independently from their antioxidant properties, the activities of these molecules have been widely investigated and described. The beneficial effects of inositols on sperm motility and mitochondrial function can be due to their many actions: other than antioxidant properties, insulin-sensitizing properties, prokinetic activity and hormonal regulatory effects. MI plays a pivotal role in reproductive physiology, positively influencing the development of oocytes, spermatozoa, and embryos. On the contrary, DCI acts to a lesser extent, having smaller effects on spermatozoa than MI. The last studies showed that DCI could be an interesting treatment thanks to its modulation upon aromatase expression, and consequently upon testosterone levels. As already demonstrated in a clinical trial, this may be particularly useful in conditions of hormonal disbalance associated with spermatic alterations.

## Figures and Tables

**Figure 1 antioxidants-10-01283-f001:**
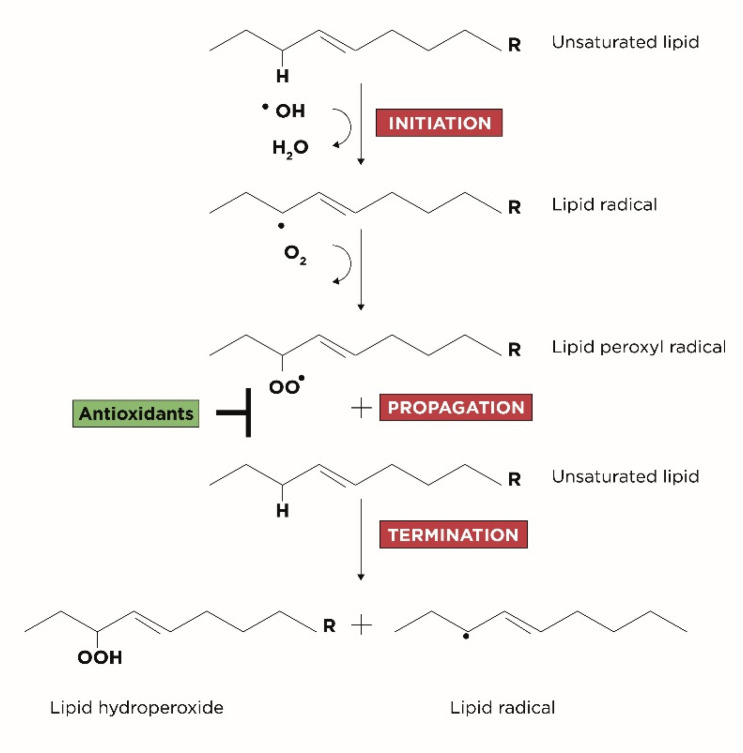
Three steps of LPO: initiation, propagation and termination.

**Table 1 antioxidants-10-01283-t001:** Myo-inositol: in vitro studies.

Author and Publication Year	Samples	Treatments	Results
Colone et al., 2010	OAT patients	Inositol 2 mg/mL and then submitted to scansion electron microscopy (SEM) and to transmission electron microscopy (TEM)	Absence of amorphous material and reduction of mitochondrial damage to the crests
Condorelli et al., 2011	5 normozoospermic and 7 OAT patients	Incubated in-vitro with 2 mg/mL of myo-inositol or placebo (control) for 2 h	Increased the number of spermatozoa with high MMP and decreased the number of those with low MMP in OAT patients
Condorelli et al., 2012	20 normozoospermic and 20 OAT patients	Incubated in vitro with 2 mg/mL of myo-inositol or phosphate-buffered saline as a control for 2 h	Increased sperm motility and the number of spermatozoa after swim-up and in OAT patients, the improvement was associated with sperm mitochondrial function.
Rubino et al., 2015	Myo-inositol group (*n* = 262 oocytes), placebo group (*n* = 238 oocytes)	Washed and subjected to swim-up with 2 mg/mL of myo-inositol or placebo-supplemented medium for 30–60 min. Spermatozoa recovered used for ICSI.	Improved spermatozoa motility in swim-up selected samples, fertilization rate (%), grade A embryos on day 3.
Artini et al., 2017	31 normospermic e 32 OAT patients	2 mg/mL MI and incubated 30 min at 37 °C	Improved total motile sperm concentration, progressive motile sperm concentration.
Scarselli et al., 2016	30 patients with grade II and III varicocele	Semen centrifuged at 1800 rpm/10 min, resuspended, and incubated with 2 mg/mL myo-inositol and 133 mg/mL myo-inositol in 9 mg/mL sodium chloride) for 15 min at 37 °C	Patients suffering from varicocele response in >60% of the samples
Palmieri et al., 2017	46 normospermic, 19 oligospermic, 15 asthenospermic patients	Semen supplemented with 15 µL/mL of myo-inositol incubated 15 min at 37 °C	Improved progressive and total motility
Mohammadi et al., 2019	40 normospermic patients	Semen divided into two aliquots ad cryopreserved: one with 2 mg/mL myo-inositol; one without myo-inositol (control)	Improved progressive and total motility, normal sperm morphology, reactive oxygen species, malondialdehyde, total antioxidant assay and DNA fragmentation
Saleh et al., 2018	41 samples: 15 normal and 26 abnormal	Semen samples supplemented with 1 mg myo-inositol to cryoprotectant	Total and progressive motility, cryo-Survival Rate
Pallotti et al., 2019	9 normokinetic semen samples with nonlinear progressive motility	Incubation with a solution of myo-inositol	Increased linear progressive motility, significant reduction in nonlinear progressive motility, increased curvilinear velocity
Governini et al., 2020	56 Caucasian males with possible causes of male infertility such as varicocele, cryptorchidism, endocrine disorders or systemic diseases	The aliquots were incubated with standard medium (untreated sample) or medium supplemented with myo-inositol at 20 mg/mL (treated sample) for 20 min.	Increase in sperm motility and in oxygen consumption, the main index of oxidative phosphorylation efficiency and ATP production, both in basal and in in vitro capacitated samples.

**Table 2 antioxidants-10-01283-t002:** Myo-inositol: in vivo studies.

Author	Study Design and Patients	Treatments	Results
Calogero et al., 2015	Double-blind, randomized, place-bo-controlled; 194 men with idiopathic infertility	Group 1 (*n* = 98) received 2 g of myo-inositol and 200 mcg of folic acid twice daily. Group 2 (*n* = 96) received one placebo sachet twice day for 3 months	MI significantly increased the percentage of acrosome-reacted spermatozoa, sperm concentration, and total count and progressive motility. In addition, reduced serum luteinizing hormone, follicle-stimulating hormone, and in-creased inhibin B concentration
Gulino et al., 2013	Prospective study; 62 patients divided into three different groups: healthy fertile patients (Group A); patients with oligoasthenospermia (OA)–(Group B)–control group (CTR).	4000 mg/die of MI and 400 µg of folic acid for 2 months	Increase of basal and after density-gradient separation method spermatozoa concentration in Group B, and a significant increase of spermatozoa count after density-gradient separation method in Group A
Montanino Oliva et al., 2016	Prospective longitudinal study; 45 asthenospermic males	The patients were treated by a dietary supplement administered twice a day containing 1 g MI, 30 mg L-carnitine, L-arginine and vit-amin E, 55 μg selenium, and 200 μg folic acid	Improved spermatic, hormonal and metabolic parameters: HOMA index, SHBG, E2, LH, free and total testosterone, sperm concentration, motility and normal morphology
Dinkova et al. et al., 2017	Prospective longitudinal study; 109 patients with astheno-zoospermia	1 g myo-inositol, 30 mg of L-carnitine, L-arginine, and vitamin E, 55 mcg of selenium, and 200 mcg of folic acid twice a day for 3 months	A significant improvement in spermmotility was reported in 85.32% of the patients

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
