# Peer review of "Oxidative Stress and Male Fertility: Role of Antioxidants and Inositols"

_antioxidants, 2021, doi:10.3390/antiox10081283_

Round 1

Reviewer 1 Report

The manuscript of De Luca et al., entitled: 'Oxidative stress and male fertility: role of antioxidants and inositols', concerns an interesting topic of ROS influence on human sperm quality, including possible role of inositols in reducing negative ROS impact.

Authors describe point by point various factors/elements related to oxidative stress in semen, their role and function. Also, the literature review concerning studies of inositols' role in sperm is done and presented in clear tables. All the construction of the manuscript is clear for the reader.

Thus, I do not have any points that should be improved.

Author Response

Dear reviewer, 

we would to thank you very much for reviewing and accepting this article review in this form.

Yours sincerely

Annarita Stringaro

Reviewer 2 Report

The authors tried to explain the correlation between male infertility and antioxidants. The issue is known from one hand but very popular form the other hand. Although it has been published a lot of similar reviews the present review adds several issues in the topic.

In the Introduction section the authors should add references about the genetic, enviromental and lifestyle factors and male fertility. The authors may wish to use the below references: Int J Environ Res Public Health. 2018 May 30;15(6):1117

In order to be more attractive to readers the authors may wish to add a paragraph about the role of viruses and male fertility, since viruses promote oxidative stress and lead to male infertility (the authors may elaborate about SARS-CoV-2). For help the authors may use the below references: Reprod Fertil Dev. 2021 Jul 30, Reprod Sci. 2021 Jan;28(1):23-26

The antioxidant therapy is a potential therapy but not always is recommendable and does not always have effects. The authors may elaborate in the Discussion section about this

Author Response

Reviewer 2

We would like to thank  for your helpful comments. We updated the manuscript as suggested. Please find below our response.

  • In the Introduction section the authors should add references about the genetic, environmental and lifestyle factors and male fertility.

-We added references on the subjects, as suggested by the Reviewer (line 39).

  • The authors may wish to use the below references: Int J Environ Res Public Health. 2018 May 30;15(6):1117.

-The aforementioned reference has been added to support the statement that exposure to chemicals represents a risk factor for male infertility (line 39).

  • In order to be more attractive to readers the authors may wish to add a paragraph about the role of viruses and male fertility, since viruses promote oxidative stress and lead to male infertility (the authors may elaborate about SARS-CoV-2).

-As suggested, we added a paragraph on the impact of viruses on male fertility, with a focus on SARS-CoV2 (lines 43-51).

  • For help the authors may use the below references: Reprod Fertil Dev. 2021 Jul 30, Reprod Sci. 2021 Jan;28(1):23-26.

-We included these as new references 11 and 13.

  • The antioxidant therapy is a potential therapy but not always is recommendable and does not always have effects. The authors may elaborate in the Discussion section about this.

-As suggested, in the conclusion paragraph we added several lines about the possible inefficacy of these treatments, also reaffirming their possible preventive role.

-Moreover, as a consequence of the new references added (7-13) all the other reference number changed. We also removed the old reference 158, which was mistakenly included but is not pertinent.

We highlighted all the change made in the text to ease the review.

Best regards

On behalf of all the authors,

Annarita Stringaro